

# The vertical Structure and spatial Variability of lower tropospheric Water Vapor and Clouds in the Trades

Ann Kristin Naumann[1,2] and Christoph Kiemle[3]

[1]Max Planck Institute for Meteorology, Hamburg, Germany
[2]Meteorological Institute, Center for Earth System Research and Sustainability (CEN), Universität Hamburg, Hamburg, Germany
[3]Deutsches Zentrum für Luft- und Raumfahrt, Institute of Atmospheric Physics, Oberpfaffenhofen, Germany

**Correspondence:** Ann Kristin Naumann (ann-kristin.naumann@mpimet.mpg.de)

**Abstract.** Horizontal and vertical variability of water vapor is omnipresent in the tropics but its interaction with cloudiness poses challenges for weather and climate models. In this study we compare airborne lidar measurements from a summer and a winter field campaign in the tropical Atlantic with high-resolution simulations to analyse the water vapor distributions in the trade wind regime, its covariation with cloudiness and their representation in simulations. Across model grid spacing from
300 m to 2.5 km, the simulations show good skill in reproducing the water vapor distribution in the trades as measured by the lidar. An exception to this is a pronounced moist model bias at the top of the shallow cumulus layer in the dry winter season which is accompanied by a too weak humidity inversion at the cloud top. The model's underestimation of water vapor variability in the cloud and subcloud layer occurs in both seasons but is less pronounced. Despite the model's insensitivity to resolution from hecto- to kilometer scale for the distribution of water vapor, cloud fraction decreases strongly with increasing
model resolution and is not converged at hectometer grid spacing. The observed cloud deepening with increasing water vapor path is captured well across model resolution but the concurrent transition from cloud-free to low cloud fraction is better represented at hectometer resolution. In particular, in the wet summer season the simulations with kilometer-scale resolution overestimate the observed cloud fraction near the inversion but lack condensate near the observed cloud base. This illustrates how a model's ability to properly capture the water vapor distribution does not need to translate into an adequate representation
of shallow cumulus clouds that live at the tail of the water vapor distribution.

## 1 Introduction

Globally moisture fields, unlike temperature fields, are not smooth but they vary on the regional scale in particular in the lower troposphere where water vapor values can be large. The distribution of water vapor strongly interacts with the atmospheric circulation through the formation of clouds and convection and through radiation. This interplay has been studied in the
tropics at the large scale (e.g., Pierrehumbert, 1995) but is less well understood in the lower tropical troposphere, where humidity is less well quantified from observations (Nehrir et al., 2017; Stevens et al., 2017). One way to fill this gap are airborne measurements taken during dedicated field campaigns. In this study, we use airborne lidar measurements from two



field campaigns in the northern tropical Atlantic to analyse the vertical structure and the spatial variability of water vapor and clouds and their representation in simulations with resolution from hecto- to kilometer scale.

Water vapor has multiple roles in the atmosphere and is closely connected to cloudiness: The boundary layer humidity sets the potential for deep convection and determines cloud amount (e.g., Keil et al., 2008; Vial et al., 2017). As the vertically integrated amount of water vapor approaches its saturation value over the tropical oceans, precipitation sets in and the amount of precipitation correlates well with the decrease in subsaturation in the column (Bretherton et al., 2004; Holloway and Neelin, 2009; Nuijens et al., 2009). On a process level, the vertical distribution of moisture determines the amount and distribution of

radiative cooling and can thereby drive large-scale and meso-scale circulations (e.g., Pierrehumbert, 1995; Muller and Bony, 2015; Naumann et al., 2019). Also, the humidity of cloud-free air in the vicinity of a cloud determines the strength of dilution of in-cloud water by entrainment. The strength of this dilution is a long-standing problem in convective parameterizations, a key ingredient of the thermostat and the iris hypothesis, and a popular tuning parameter (Ramanathan and Collins, 1991; Mauritsen et al., 2012; Mauritsen and Stevens, 2015).

The vertical distribution of moisture and small-scale phenomena such as the dilution of clouds by entrainment are posing challenges to both modelling and observations. The WALES (WAter vapor Lidar Experiment in Space) lidar is capable of profiling moisture, aerosol, and clouds simultaneously with high accuracy and spatial resolution (Wirth et al., 2009). High resolution in vertical profiles is of particular importance in the tropics since sharp moisture gradients at the trade inversion influence radiation locally (Stevens et al., 2017). Installed on an aircraft, measurements with WALES can be undertaken in

regions of particular interest. In December 2013 and in August 2016 the NARVAL (Next-generation Aircraft Remote-sensing for VALidation) campaigns were the first tropical experiments in which an airborne water vapor lidar participated (Stevens et al., 2019b). For the two campaigns the German research aircraft HALO (High-Altitude Long-Range) sampled the western tropical Atlantic East of Barbados to investigate the interactions between shallow moist convection, moisture distribution, and radiative effects with a state-of-the-art suite of remote sensing instruments and dropsondes.

The close coupling between clouds and water vapor and the capabilities of lidar measurements in the trade wind regime motivate the guiding questions of this study: What is the vertical structure and the spatial variability of water vapor in the trades? How does cloudiness covary with water vapor and are models able to represent the observed relationship correctly?

In numerical weather prediction, storm resolving model (SRM) simulations with kilometer-scale grid spacing are common and evaluated frequently (e.g., Bauer et al., 2015). Aiming to better resolve convection with higher resolution, traditional

idealized large-eddy model (LEM) simulations lack the ability to represent the mesoscale and large-scale variability of observed cloud fields (Nuijens and Siebesma, 2019). LEM simulations with hectometer scale grid spacing are now becoming available on large domains with realistic boundary conditions (Heinze et al., 2017; Stevens et al., 2019b). These LEM simulations with realistic and varying large-scale states include the interaction with the large-scale circulation and at the same time the subgrid-scale flow is better constrained than in coarse resolution simulations. Although simulations with hectometer grid spacing still

do not have a grid spacing fine enough to represent details of shallow convection, even kilometer-scale simulations are found to reproduce many features, such as the daily cycle in cloud amount and precipitation, better than climate models with convective parameterization (Stevens et al., 2019a; Vial et al., 2019). It is an open question whether hecto- and kilometer-scale simulations





with realistic and varying large-scale states are able to represent water vapor variability and its co-variation with clouds in the trades and whether this ability depends on resolution.

In model simulations convection, due to its stochastic nature, is not expected to trigger in the exact same position and with the exact same timing as in reality. Therefore comparisons between observations from line-shaped research flights and models, where the comparison is based on co-location of the two in space and time, are often of limited use. To bypass the issue of co-location other means of comparison are needed. We propose to compare model and observations in moisture space, i.e., we sort water vapor profiles from the driest to the wettest profile, to identify differences in the vertical structure of water vapor and

its change in moisture space. The depiction of humidity in moisture space is inspired by Bretherton et al. (2005), who compare model results as a function of column-relative-humidity to illustrate the mechanisms of convective self-aggregation in radiative convective equilibrium. In observations this technique has been first used by Schulz and Stevens (2018). With a comparison of observations and simulations in moisture space we avoid relying on co-location but retain the ability to quantify variability at high spatial resolution.

This paper is organized as follows: Section 2 describes the observations and model simulations used in this paper. In Sect. 3 we focus on the case study of a research flight on 11. December 2013, which is a case of typical shallow trade wind convection and is also used to explain our methodology in detail. In Sect. 4 we generalize the results of the case study by applying the same methodology to a set of research flights that allow us to analyse the seasonality of the water vapor structure in the trades. Conclusions are given in Sect. 5.

## 75  2   Observations and Model Simulations

### 2.1   NARVAL winter and summer campaign

Two NARVAL field studies took place over the tropical Atlantic ocean east of Barbados (Stevens et al., 2019b). The first part of the field study counts eight research flights between 10 - 20 December 2013 and the second part ten flights between 8 - 30 August 2016. The details of the NARVAL field studies, such as the flight strategy and the description of the HALO aircraft are

described by Stevens et al. (2019b) and Konow et al. (2019). Not all data are to the same degree useful for this analysis, as some of the long flights (e.g., the transit flights between Germany and Barbados) are not contained in the modeling domain of the LES (see Sect. 2.3) and some other days have not been chosen to be modeled with LES. For the purpose of this study, we limit the available lidar and microwave radiometer data by the criterium of being included in our smallest modeling domain (see Section 2.3). The time and domain constraints are given in Table 1.

Basic differences between the winter and the summer trades appear in the cloud layer moisture and thickness (Table 1). While the winter situations are characterised by similar and undisturbed trade wind conditions, the summer flights encountered a significant layer of Saharan dust on August 12 and 19, the flight on August 22 was close to the intertropical convergence zone (ITCZ), and the flight on August 24 was close to the tropical storm Garcon (Gutleben et al., 2019).





**Table 1.** Specification of flight domains used in this study.

| $t$ in UTC | domain | $N$ | $p$ in % | $q_c$ in $\mathrm{g\,kg}^{-1}$ | $h_c$ in km |
|---|---|---|---|---|---|
| *NARVAL 1* | | | | | |
| 11. Dec 2013 16 - 21 | 10.0 - 16.5 N, 58.0 - 55.0 W | 531 | 34.2 | 4.0 | 3.0 |
| 12. Dec 2013 14-15, 19-20 | 14.0 - 16.5 N, 56.5 - 48.5 W | 526 | 86.5 | 4.0 | 2.8 |
| 14. Dec 2013 14-15, 19-20 | 13.9 - 16.5 N, 57.2 - 48.5 W | 296 | 48.9 | 4.0 | 2.5 |
| 15. Dec 2013 16 - 21 | 12.0 - 16.5 N, 57.5 - 48.5 W | 668 | 72.8 | 4.0 | 2.7 |
| 20. Dec 2013 17 - 18 | 13.3 - 16.5 N, 56.0 - 51.6 W | 168 | 70.3 | 4.0 | 3.0 |
| *NARVAL 2* | | | | | |
| 12. Aug 2016 13 - 19 | 9.5 - 14.0 N, 55.0 - 52.0 W | 1317 | 69.0 | 6.0 | 1.9 |
| 19. Aug 2016 13 - 17, 20 | 13.5 - 16.0 N, 57.0 - 48.0 W | 1115 | 85.4 | 8.0 | 2.6 |
| 22. Aug 2016 14-15, 20-21 | 10.0 - 12.8 N, 58.6 - 51.0 W | 279 | 55.9 | 8.0 | 1.8 |
| 24. Aug 2016 13 - 16 | 13.0 - 14.5 N, 56.5 - 44.0 W | 405 | 51.3 | 9.0 | 1.6 |

$t$: time period of analyzed flight, $N$: number of valid profiles, $p$: percentage of valid profiles, $q_c$: water vapor mixing ratio threshold for detecting a cloud top with WALES, $h_c$: maximum shallow cloud top altitude

## 2.2 WALES Lidar and HAMP Radiometer

The differential absorption lidar WALES is installed pointing downwards on the HALO aircraft, measuring water vapor profiles throughout the tropical troposphere with three on-line laser wavelength positions in the near-infrared situated on three water vapor absorption lines of cascading strength (Wirth et al., 2009; Kiemle et al., 2017; Gutleben et al., 2019). The weakest line, specially selected for the tropics, permits accurate profiling of very moist layers below the trade inversion while the stronger two lines provide reliable data of the moisture jump across the inversion and the dry regions above. Backscatter from aerosol

and clouds, corrected for extinction by aerosol, is simultaneously measured by a high spectral resolution lidar (HSRL) at 532 nm with a temporal resolution of 1 s, corresponding to a spatial horizontal resolution along the flight route of 210 m given the typical aircraft speed of $210\,\mathrm{m\,s}^{-1}$ during the summer campaign and a horizontal resolution of 240 m given an aircraft speed of $240\,\mathrm{m\,s}^{-1}$ during the winter campaign. Flight speed was higher in winter due to a higher average flight altitude. To achieve an acceptable measurement precision of typically 10 % in the cloud layer and above, the water vapor profiles are aggregated

across 12 s or about 2.5 km in the summer campaign and 2.9 km in the winter campaign. The vertical resolution is about 250 m for water vapor and 15 m for backscatter. Water clouds quickly attenuate the lidar signal such that valid data are only obtained above cloud top clearly defined by the backscatter signals (Fig. 1 a). Full profiles are obtained wherever the cloud gaps are larger than 2.5 km.

Since our focus is the cloud layer moisture variability we only use those lidar profiles where more than half of the data

points below the maximum cloud top height, which is defined by $q_c$ in Table 1, are valid. For example, on 11 December 2013, the cloud layer top height is 3.0 km, and only in 34 % of all lidar profiles more than half of the data points are valid





below this height (Fig. 1 a). The rest is unavailable due to clouds or laser adjustment phases. We consequently use only one third of all profiles of this flight (Fig. 1 b). This subset still contains small gaps mainly due to clouds which we fill with the saturation value by assuming saturation wherever the HSRL backscatter coefficient is $> 10 \ (\mathrm{Mm\,sr})^{-1}$ which to sufficient

approximation defines a water cloud (Kiemle et al., 2017). We deviate from this threshold only in two cases where the clouds are particularly small (on 12 August 2016 we use $5 \ (\mathrm{Mm\,sr})^{-1}$ to compensate for the signal dilution) or large (on 24 August 2016 we use $15 \ (\mathrm{Mm\,sr})^{-1}$). We fill the remaining gaps with the moisture of the nearest neighbor profile in the horizontal and call this gap-free result a minimum estimate (WALES$_{\mathrm{min}}$; Fig. 1 c). In a maximum estimate (WALES$_{\mathrm{max}}$) we additionally fill all original cloud shadows down to the lifting condensation level (LCL), i.e., missing data below lidar-detected clouds, with

the saturation value. We use the lidar signals from thin boundary layer clouds to find the LCL and temperature profiles from close dropsondes to determine the saturation humidity profile. Since the thickness of the cloud cannot be determined by the lidar and also lower clouds may exist above the LCL, the maximum estimate gives an upper bound on cloudiness and water vapor path (WVP, defined as the vertically integrated specific humidity without contributions from liquid or ice). Likewise the minimum estimate provides a lower bound on cloudiness and WVP. Consequently, the difference between the minimum

and the maximum estimates characterises to a satisfying extent the uncertainty of our attempt to quantify the lidar moisture distribution within and below the clouds while aiming to obtain a gap-free data curtain needed for the model comparisons. We will show later that the uncertainty in the measured humidity estimate is small compared to the difference between model and observation (see Section 3.2). To obtain the cloud fraction in the 12-s grid, we apply the abovementioned HSRL backscatter coefficient threshold for water clouds onto the 1-s lidar backscatter curtains, using a similar min/max assumption to account

for measurement and methodical uncertainties.

To understand which part of the moisture space the WALES lidar misses in cloudy environments, we additionally make use of the HAMP (HALO Microwave Package) radiometers, whose data is available for NARVAL 1 (Jacob et al., 2019a) and NARVAL 2 (Jacob et al., 2019b). The nadir-viewing HAMP microwave radiometers measure the WVP with 1 s (that is 210 m or 240 m) resolution along the HALO flight track (Jacob et al., 2019c). Their co-alignment with the lidar field of view was

checked by comparing the radiometer liquid water path with the lidar cloud backscatter signals, both available at 1 s resolution. The radiometer signals are interrupted by calibration events. Comparisons with the co-located lidar WVP reveal that those events are independent from the ambient humidity conditions. The radiometer WVP distributions are consequently not biased, except for a slight underrepresentation of the moistest scenes due to saturation which concerns less than 1.5 % of all WVP data.

## 2.3 ICON

Simulations are run with ICON (Icosahedral non-hydrostatic model; Zängl et al., 2015) with four different grid spacings between 2.5 km and 300 m and with two different model versions: ICON-SRM and ICON-LEM. The ICON-SRM was run with 75 vertical levels and with 2.5 km and 1.25 km horizontal grid spacing. Details of the simulations are described by Klocke et al. (2017). The ICON-LEM (Dipankar et al., 2015; Heinze et al., 2017) was run with 150 vertical levels and with

600 m and 300 m horizontal grid spacing. Details of the simulations are described by Stevens et al. (2019b). In all simulations





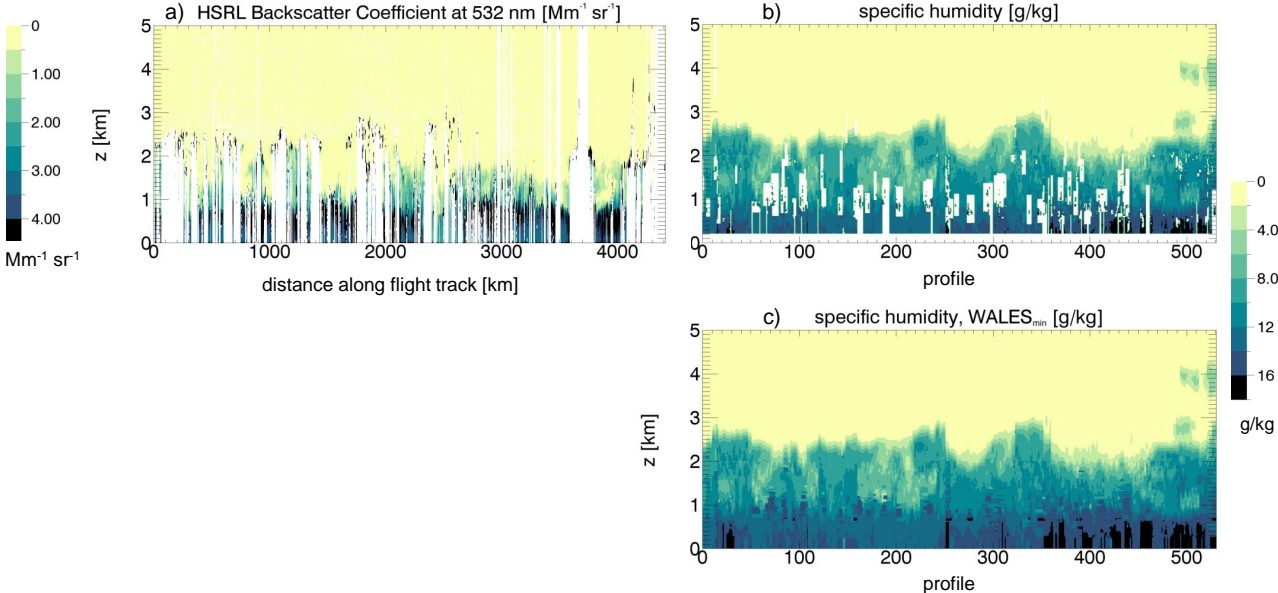

**Figure 1.** Lidar profiles of the flight on 11. December 2013: (a) atmospheric backscatter for the full flight, and (b,c) specific humidity for those 531 profiles where more than 50 % of the lidar data within the cloud layer and below are available. The remaining gaps in the original data set in (b) are filled by assuming saturation in clouds, and by nearest neighbor values elsewhere, resulting in a gap-filled representation in (c). See text for details. Note that the aspect ratio is 1:500 in (a) and 1:150 in (b) and (c).

the parameterizations for shallow and deep convection, gravity wave drag and subgrid-scale orography are switched off. The parameterizations for turbulence and microphysics differ between the SRM and the LEM. In addition, the SRM simulations apply a cloud cover parameterization while the LEM simulations use a binary approach. For this study, we set the LEM cloud fraction to 1 if the liquid water content in a grid box is non-zero, and 0 otherwise.

The SRM runs with the coarsest grid spacing of 2.5 km cover the largest domain including the entire tropical Atlantic (10.0 S - 20.0 N, 68.0 W - 15.0 E). The simulated domain size decreases with increasing resolution, so that the LEM run with the finest grid spacing of 300 m has the smallest domain, which still covers an area of 800 km × 1600 km in the western part of the Atlantic (8.0 - 16.5 N, 60.0 - 43.5 W). For the purpose of this study, we do not analyze model output from the full simulation domains of ICON at different resolutions but instead limit the domain analyzed to rectangles around those parts of the flight

paths that took place within the smallest simulated domain. Because the flight paths and time periods differ from day to day, the analyzed domains and time periods also differ as given in Table 1.

Initial and boundary conditions for the ICON-SRM 2.5 km simulations are taken from the European Centre for Medium-Range Weather Forecast (ECMWF) reanalysis and vary in time except for the SST, which is fixed for each simulation day. The simulations apply a one-way nesting of higher resolution simulations in low resolution simulations. The ICON-SRM

simulations with 2.5 km horizontal grid spacing apply an online refinement to 1.25 km via nesting in the eastern part of the domain and start at 0 UTC for each day of December 2013 and August 2016. They are run forward in time for 36 hours.





ICON-LEM simulations are initialized and nudged at the lateral boundaries from ICON-SRM and start at 9 UTC for selected days to match the flight operations of the NARVAL campaign. They are run forward in time for 27 hours. Simulations are analyzed from hourly model output starting earliest at 13 UTC (see Table 1) so that a sufficient spinup period is taken into
account.

## 3   Case study: Covariation of clouds and moisture

In this section, we use one day of the first NARVAL campaign, 11. December 2013, to explain our methodology of comparing lidar measurements with model results in moisture space and to illustrate some prominent features of covariation of clouds and moisture (Fig. 1 and Fig. 2).

### 3.1   Synoptic Situation and Flight

We choose the 11. December 2013 for a detailed case study for two reasons: First, a regular meander flight pattern allows us to sample a well-defined region thoroughly, which aids a comparisons with simulations (Fig. 2). Second, the conditions seem preferential to sample the humidity space because the flight area includes typical shallow convection over most of the area but also approaches deeper convection with higher humidity towards the south.

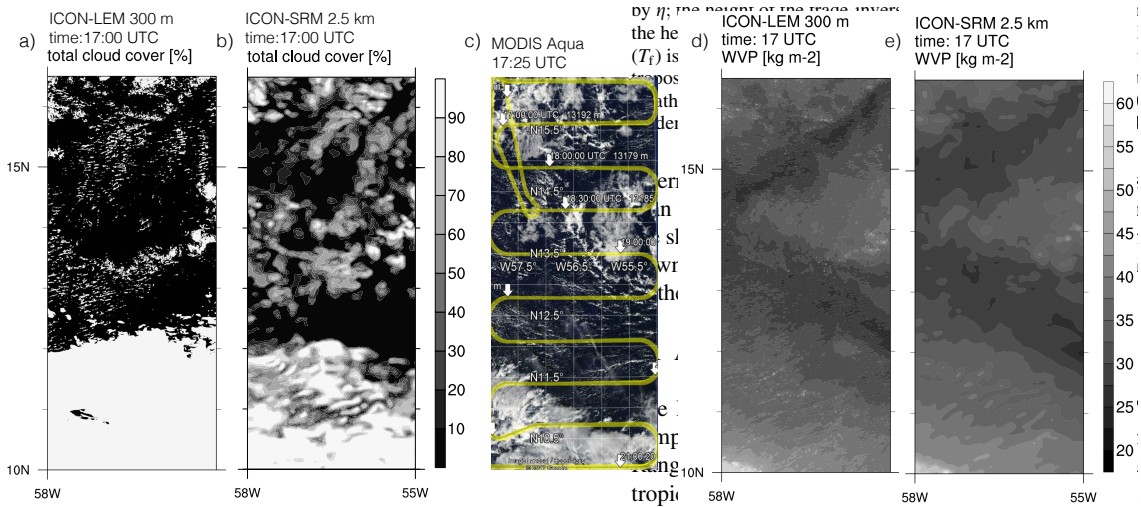

**Figure 2.** Cloud cover and water vapor path (WVP) in the flight domain on 11 December 2013. Cloud cover at 17:00 UTC from (a) ICON-LEM 300 m and (b) ICON-SRM 2.5 km; (c) MODIS Aqua corrected reflectance at 17:25 UTC overlaid with the flight path which was flown from north to south; WVP at 17:00 UTC from (d) ICON-LEM 300 m and (e) ICON-SRM 2.5 km.

The modeled cloud structures have similarities with the observed reflectance from MODIS showing organized structures of shallow clouds in the northern three quarters of the domain (Fig. 2). With a grid spacing of 2.5 km these shallow clouds have a too broad structure compared to observations. With higher resolution the cloud structures, not surprisingly, become finer but at





300 m grid spacing the model misses some stratiform outflow from shallow cumulus giving the shallow convective cloud field a less organized appearance than in satellite observations. In both simulations and in the satellite view the southern quarter

of the domain is dominated by a cirrus shield originating from deep convection just south of the domain. This cirrus shield is reaching further north in the model than in the satellite observations. Because the deep convective system moves towards the south west with time and the flight itinerary is following the pattern from north to south, the lidar observations onboard the aircraft catch only a small amount of this regime (see Sect. 3.2).

The field of WVP shows more small-scale structure at 300 m grid spacing than with 2.5 km but changes less with resolution

than the cloud cover does. All simulation show an increase of WVP from north to south and a c-shape of low WVP in the northern and central section of the domain. This c-shape in the modeled WVP can be surmised in a reduced presence of clouds in the satellite view but is less well reflected in the modeled cloud cover.

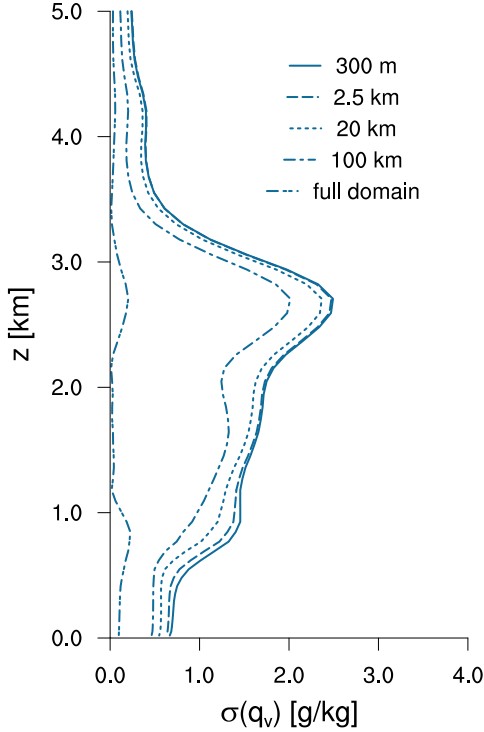

**Figure 3.** Contribution of different scales to the standard deviation of $q_v$ on 11 December 2013 from ICON-LEM 300 m. Domain and temporal coverage are given in Table 1. Simulations with a grid spacing of 300 m have been coarsend to squares with side lengths of 2.5 km, 20 km, 100 km, and "full domain", which corresponds to a side length of about 400 km. Both spatial and temporal variability contribute to the standard deviation except for the "full domain", which only shows temporal variability. The cloud layer ranges from $z = 0.5$ km to $z = 3.0$ km.

Averaging the results of the ICON-LEM 300 m simulation on squares of different side length, we analyse how the standard deviation of the water vapor mixing ratio, $q_v$, changes with effective resolution (Fig. 3). Coarse graining the 300-m LEM results





to 2.5 km does not change the standard deviation considerably. The relative contribution of small scales between 300 m and 2.5 km to the standard deviation of $q_v$ is largest near cloud base and in the subcloud layer but generally well below 10 %. Even for a side length of 20 km the relative differences to the native grid spacing of 300 m are maximum near cloud base (30 %) but are considerably smaller throughout the cloud layer and above (< 10 %). Because the differences are small, for the remainder of this analysis we show model results and observational data at their native scale (from 300 m to 2.5 km), which aids a direct

evaluation of what a simulation is able to catch without artificially reducing information by averaging.

### 3.2 Spanning the Moisture Space

Because of its stochastic nature convection is not expected to trigger at the exact same location and time in simulations as it does in reality. To bypass the issue of co-location, we sort water vapor profiles from the driest to the wettest profile and compare simulations and observations in moisture space (Bretherton et al., 2005; Schulz and Stevens, 2018). Comparing

simulation results with data from HAMP, this procedure is straight forward because the HAMP dataset samples the whole domain well. WALES on the other hand is rapidly attenuated in clouds and saturated in the wettest profiles so that a fair comparison to simulations needs to take into account information on which situations WALES is not able to observe. We therefore use HAMP to span the moisture space, to quantify what WALES misses, in particular in the wet regions, and to construct a "stretched moisture space" that enables a fair comparison between WALES and ICON. This method works well

during NARVAL because flight patterns were fixed before takeoff and hence measurements along the flight path represent a random sample of the encountered cloud regime. The validity of this method quickly reaches its limits if flight paths are adjusted to preferentially sample a feature of special interest – a trade-off to be aware of for future flight planning (e.g., in view of EUREC[4]A, Bony et al., 2017).

We proceed as follows: All available WVP values from HAMP and the ICON simulations at different resolution are sorted

from the lowest to the highest value (Fig. 4 a). This representation corresponds to the cumulative distribution function of WVP, which is rotated by 90° compared to the common depiction. Since WALES and HAMP measure the same location at the same time, a co-location between those two instruments is eligible. WALES_min values scatter around HAMP values with a standard deviation of $1.48 \, \mathrm{kg/m^2}$ ($1.62 \, \mathrm{kg/m^2}$ for WALES_max), which is consistent with Jacob et al. (2019c). Because WALES measurements attenuate quickly in clouds and for high WVP (Sect. 2), data gaps are not randomly distributed in moisture space

but instead preferentially occur where WVP is high: of the driest 10 % of HAMP measurements 48 % have a corresponding measurement from WALES, while for the moistest 10 % of HAMP measurements only 3 % have a corresponding measurement in WALES. To account for this biased sampling of WALES, we randomly select model results and HAMP according to these percentages of WALES counterparts in each 10 % interval. Then we sort all WALES WVP by its increasing value. The resulting new moisture space of all valid WALES data points and those subsampled from ICON and HAMP is effectively stretched in

its drier part and compressed in the moister part (Fig. 4 b and lower x axis in Fig. 4 a). We call this new moisture space the stretched WVP space according to WALES or, in short, the "stretched moisture space". This stretched moisture space enables a fair comparison between WALES and ICON.





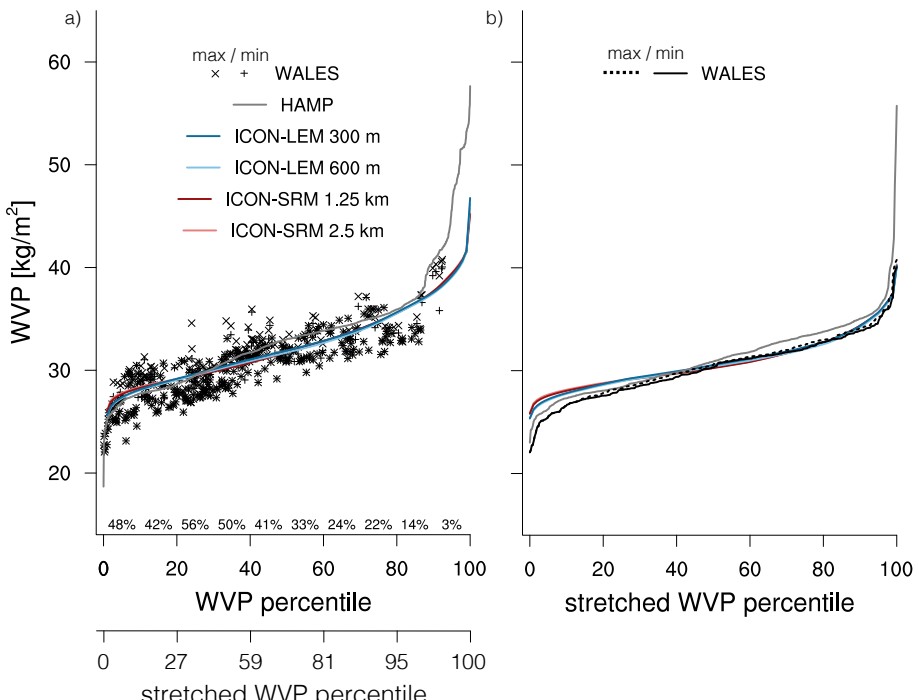

**Figure 4.** Water vapor path (WVP) on 11th of December 2013 in WVP space. (a) ICON simulations and HAMP observations of WVP are sorted by each one's WVP values. WALES data is plotted as co-located with HAMP. Percentages above the x axis tell how many valid WALES measurements have been obtained in each 10 % interval of HAMP's WVP space. (b) ICON results and HAMP data is randomly selected according to those percentages in each 10 % interval resulting in a stretched WVP space, which is also shown as an additional x axis in (a). In (b) WALES data is sorted by its own WVP instead of being co-located with HAMP. Further details are discussed in the text.

In stretched moisture space, the distribution of WVP from ICON simulation results, and WALES and HAMP measurements overall agree well (Fig. 4 b). The differences between the three observational estimates, HAMP, WALES$_{min}$, and WALES$_{max}$ are small with a median of absolute difference around 0.6 kg/m$^2$ (WALES$_{min}$ vs. HAMP: 0.60 kg/m$^2$, WALES$_{max}$ vs. HAMP: 0.56 kg/m$^2$). The differences in the distributions of WVP between simulations at different grid spacing are much smaller. This possibly reflects the nested modeling approach, which ensures consistent initial and boundary conditions and where domains are nudged with a time scale of 3 h, to ensure that they do not deviate too much in the two-way setup. However, the differences in cloud fraction are considerably larger (see Sect. 3.3), which indicates that the effect of grid spacing in the range of hecto- to kilometer scale is small for the distribution of the WVP.

The small intra-observational and intra-model differences enable a meaningful interpretation of the difference between model and observation. Compared to observations the modeled variability of WVP is too small. The driest model areas are too wet, while the wettest model areas agree well with WALES (Fig. 4 b). This underestimation of the variability in WVP can be attributed to too low variability of moisture in the cloud layer (see Sect. 3.3). If WVP is not subsampled for valid WALES profiles, there is also a dry model bias for very wet profiles as compared to HAMP (Fig. 4 a). Here, the wettest 15 % of HAMP's





moisture space seem to be not well represented in the model. Two factors are expected to contribute to this deviation: On 11 December 2013 there is a little change in the flight track near 11 N 56 W. This was made to try to fly over the deepest turret of the towering convection and try to drop a sonde through this (personal communication, Bjorn Stevens, 2019). Hence this flight segment is purposely biased to the moistest cell and may contribute to differences in the moist part of the space of Fig. 4
a. Also, extending the analysed model domain to south of 10 N, decreases this bias which suggests that the deep convective system on 11 December 2013 is consistently placed too far south in all four simulations (not shown). Because both the deepest turret of the towering convection and in general the moistest profiles towards the south of the domain contain less valid WALES samples than the drier profiles, this feature is much less visible in stretched moisture space and is therefore less important for the remainder of this analysis.

**3.3  Vertical Distribution of Water Vapor and Cloud Fraction**

With the framework of the stretched moisture space, we can now also analyse the vertical structure of water vapor and cloud fraction by comparing valid WALES profiles with ICON profiles that are subsampled according to percentages of the WALES counterpart. The analysis therefore does not represent the real space as an omniscient observer would see it but only that part that WALES is equipped to measure.

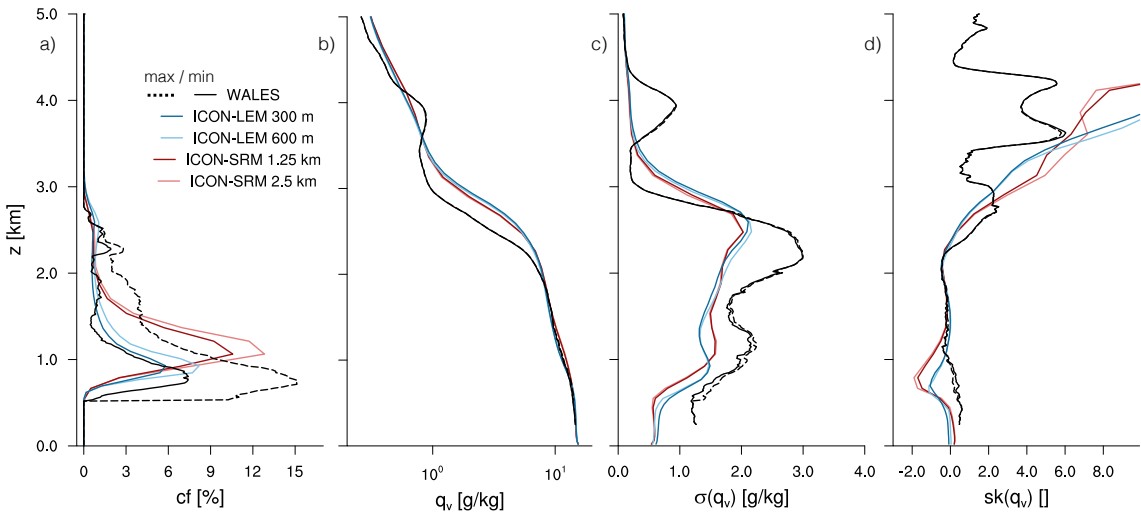

**Figure 5.** Profiles of (a) cloud fraction, (b) mean water vapor, $q_v$, and its (c) standard deviation and (d) skewness for the 11. December 2013 in stretched moisture space as defined in Fig. 4.

The mean water vapor mixing ratio compares well between WALES and ICON (Fig. 5 b). As for the integrated quantity WVP, also in the vertical structure of $q_v$ there is no dependence on grid spacing. Compared to WALES the inversion is too high in the model, a feature that is common to all analysed days in December 2013. Both the observed and the modeled heights of the inversion increase with increasing WVP but this increase is less pronounced in the simulations (Fig. 6 b). For the dry

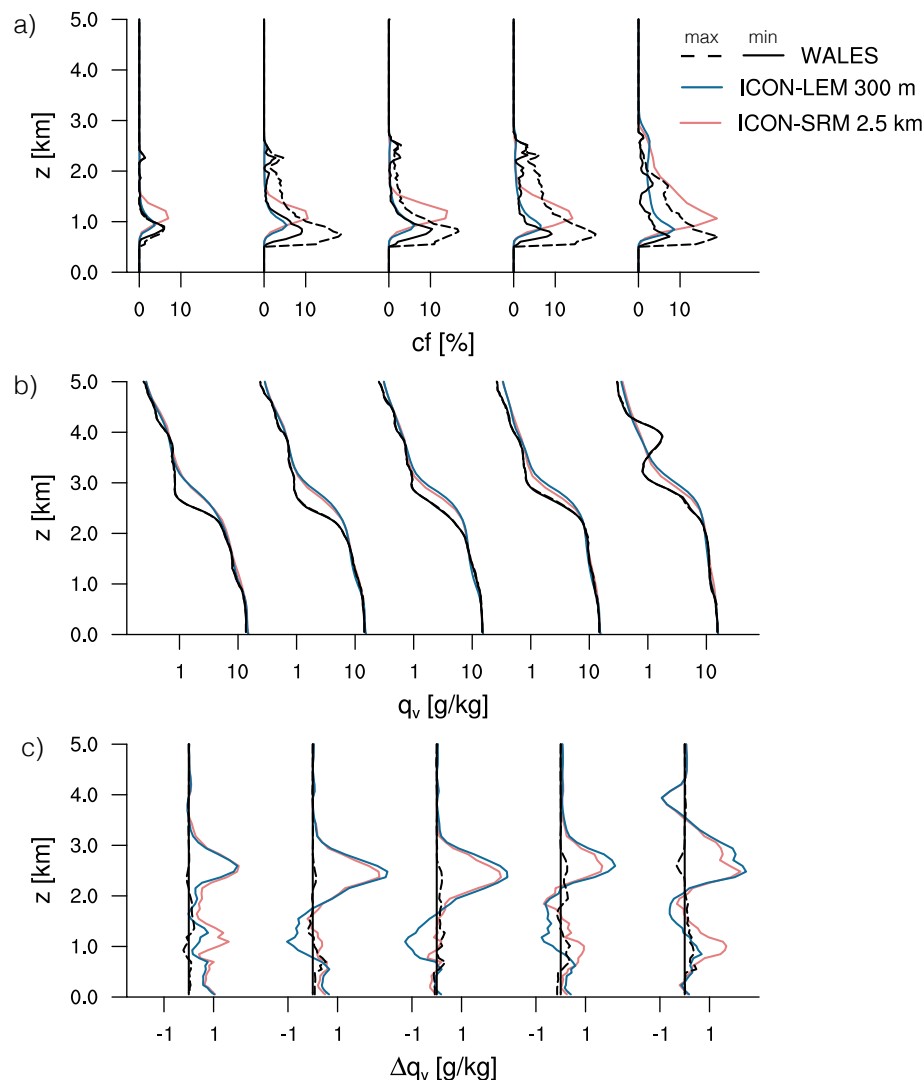

**Figure 6.** Profiles of (a) cloud fraction, (b) water vapor and (c) the difference of water vapor to the WALES$_{min}$ estimate for 11 December 2013. Each profile shows the mean for a 20-percentile range of WVP in stretched moisture space from driest profiles on the left to moistest profiles on the right (see Fig. 4). To retain fluctuations due to a limited number of samples, as many profiles as are available from WALES have been randomly subsampled from ICON results (here 531 samples, see Table 1). At a given height level WALES$_{max}$ can be lower than WALES$_{min}$ because the sorting of profiles is done according to the column integrated WVP separately for the minimum and the maximum estimate.

profiles the modeled inversion is also less steep, which implies a less concentrated radiative cooling in the simulations at the
cloud layer top with possible implications for mesoscale circulations (Naumann et al., 2019).



The higher moments of the water vapor distribution do not agree as well as the mean but still capture the main features and the right magnitude. The two maxima of the standard deviation of $q_v$ in the cloud layer are well captured but are underestimated by the model compared to the observations (Fig. 5 c). This is also evident from the change in bias with increasing WVP: in the cloud layer the driest profiles tend to be too moist in the model (Fig. 6 c).

The skewness, which is defined as the ratio of the third central moment of the distribution to the $3/2$ power of the variance, is reasonably well represented from the middle of the cloud layer up to the cloud layer top (Fig. 5 d). Near cloud base the model simulates a negative skewness, that is, few very dry locations associated with cloud free regions, while the observations indicate slightly positive values, that is, few very moist locations. This difference in sign between model and observations is also found on the 14. and 15. December 2013 but not on the other days (not shown). Above cloud top between 4 km and 7

km the modeled skewness is very large, which is caused by a single deep convective cell near the south-western corner of the domain that dominates the skewness but has not been sampled during the campaign and is therefore not represented in the observations.

While these properties are characteristic also for other flight days of the NARVAL campaign, a feature that is special to the observations on 11. December 2013 is a secondary maximum at 4 km height (Fig. 5 b). This secondary maximum is evident

only in the moistest profiles (Fig. 6 b), manifests in the southern part of the domain towards the end of the flight (Fig. 1) and is caused by a moist outflow from convectively more active regions. This feature is also reflected in higher values of standard deviation and skewness but is absent in all three moments in the model, which misses the moist outflow (Fig. 5 c, d).

For the mean cloud fraction, both uncertainties from observations and sensitivity to model resolution are larger than for $q_v$ (Fig. 5 a). From WALES the uncertainty in maximum cloud fraction is a factor of two (between 7.4 % for WALES$_{min}$

and 15.2 % for WALES$_{max}$) but the vertical structure is similar with a clear maximum in cloud fraction near cloud base and few shallow clouds deepening up to 3 km. This structure is also represented well by the simulations, except that the cloud fraction maximum is placed too high. We suspect that this upward shift in cloud fraction maximum is linked to the resolution because the shift is stronger for the SRM than the LEM simulations. Another resolution dependent feature is the value of the maximum cloud fraction, which decreases substantially by a factor of two between 12.8 % (ICON-SRM 2.5 km) and 5.8 %

(ICON-LEM 300 m) but is still included in the range of uncertainty given by the observations. Hohenegger et al. (2019) find similar dependencies of cloud fraction on grid spacing between 2.5 km and 80 km and hypothesize that if horizontal resolution is not sufficient for proper mixing, the boundary layer grows and clouds form higher at colder temperatures leading also to more cloudiness. The decrease in cloud fraction between the simulations with 600 m and 300 m grid spacing is still substantial and not converged, which is in agreement with idealized modelling studies showing that LEM underestimates cloud fraction

when the grid spacing becomes as fine as 50 m (Vogel et al., 2019).

With increasing WVP the clouds deepen from very shallow cloud tops around 1 km up to cloud tops around 3 km both in the simulations and in observations (Fig. 6 a). Whether the maximum cloud fraction also increases with increasing WVP is not clear: for WALES$_{min}$ the maximum cloud fraction stays about constant while for WALES$_{max}$ the maximum cloud fraction increases with increasing WVP. Cloud fraction from the LEM simulations agrees well with the WALES$_{min}$ estimate but in the

SRM simulations the maximum cloud fraction increases similar to the WALES$_{max}$ estimate. For features other than the height





of the maximum cloud fraction, which is shifted upward in particular in the SRM simulation, it therefore remains unclear for this case study whether the modeled cloud fraction improves with resolution or not. For the season of August 2016 a better representation of cloud fraction with higher resolution becomes apparent and will be discussed in the next section.

## 4 Seasonal Composites

In this section we generalize the results of the case study by applying the same methodology to composites of several research flights that allow us to analyse different regimes of the water vapor structure in the trades. We combine five research flights in December 2013 to one composite case and four research flights from August 2016 for another composite case (Table 1), both of which represent different seasons in the trades. As for the case study in the previous section, we subsample all model results according to the percentages available from WALES in each 10 % bin of WVP to enable a fair comparison between model 295 results and observational data. The analysis in this section is therefore discussed in the resulting stretched moisture space.

### 4.1 Stretched Moisture Space

Boreal winter in the northern trades near Barbados is generally characterized by a drier free troposphere compared to boreal summer, which is characterized by more frequent disturbances, a closer proximity of deep convection associated with the ITCZ, and a moister free troposphere (e.g., Stevens et al., 2017). All research flights in December 2013 took place in a period 300 of undisturbed shallow convection (Vial et al., 2019). To analyse whether the chosen research flights characterize a meaningful regime of water vapor structure, we test their representativeness by extending the analyzed period to the ambient days (10. to 21. December 2013) and choosing the mean borders of their domains (12.7 - 16.5 N, 57.0 - 50.4 W). For December 2013 the research flights represent the extended period very well (Fig. 7 a). For August 2016, we extend the period and domain in the same way except for the southern border (11. to 25. August 2016; 13.0 - 14.3 N, 56.8 - 48.8 W). Compared to the mean border, 305 the southern border is shifted 1.5° north to avoid inclusion of deep convection from the ITCZ on a few days, where it reaches further north. In August 2016, the extended period is several kg/m$^2$ moister than the flight period and domain. This difference can be explained by two factors: On several of the flights in August 2016 dry sectors were sought out purposely biasing the flight periods compared to the extended period (Bjorn Stevens, personal communication, 2019). This illustrates the problem of flying toward specific features, rather than fixing a flight pattern to sample a region evenly (see also Sect. 3.2; Jacob et al., 310 2019c). In addition, on 20 - 22 August 2016 the tropical cyclone Fiona runs by north of the domain and brings some very moist air into the domain behind it on 23 August 2016 contributing to a moister extended period. Because the difference between the moist August flights and the dry December flights is considerably larger than the difference between the flight periods and their extended periods, both composite cases can be seen as representative for different regimes. A good representation of the NARVAL flights for their respective season is also found by a comparison with a 8 year long time series at the Barbados cloud 315 observatory in terms of cloud depth and base (Heike Konow, personal communication, 2019).

As for the case study also in the seasonal composites of the flight domains the stretched distribution of WVP agrees well between model and observation (Fig. 7 b, c). The uncertainty in the observational estimate as well as the sensitivity to model





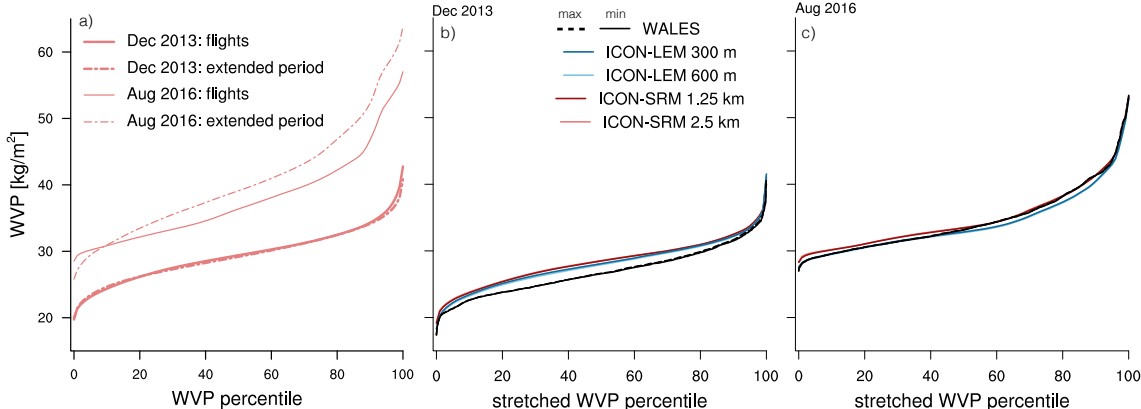

**Figure 7.** WVP as a function of WVP percentiles (a) for ICON-SRM 2.5 km and the flight period and domain in December 2013 and August 2016 (see Table 1) as well as for an extended period that includes a longer time period for a domain with mean borders (see text for details); (b) for the flight composite in December 2013; and (c) for the flight composite in August 2016.

resolution is small for both seasons. In December 2013 the model tends to be too moist with the largest bias up to 2 kg/m$^2$ between the 10th and the 60th percentile and a smaller moist bias for the very low and the high WVPs. In August 2016, the

agreement is excellent. The LEM results fall almost exactly on the WALES estimate for the lower half of the stretched moisture space and the SRM results coincide with the WALES estimate in the upper half of the stretched moisture space.

### 4.2 Vertical Distribution of Water Vapor and Cloud Fraction

For the December composite the vertical distribution of mean water vapor, its first moments, and the cloud fraction is very similar to the case study on 11. December 2013 (Sect. 3). We find good agreement between model and observation both in value

and shape of the vertical profiles with a few exceptions (Fig. 8 a-d): a moist model bias at the inversion, an underestimation of the standard deviation of $q_v$ in the cloud layer by the model, the model's negative skewness of $q_v$ at cloud base as compared to a positive value in observations, and an upward shift of the modeled height of the maximum cloud fraction. One difference to the case studies of 11. December 2013 is a stronger secondary maximum of cloud fraction near 2 km height in the simulations with 600 m to 2.5 km grid spacing. These small stratiform cloud shields below the inversion are often present in both model

and observations (Lamer et al., 2015; Vogel et al., 2019) but cannot be found in the WALES data in our time period. The LEM simulations with finest grid spacing (300 m) are closer to the observations in this case.

    Compared to the December composite the August composite is characterized by a moister free troposphere and a shallower cloud layer ($< 2$ km, Fig. 8 e-h). This supports the understanding that a moister free troposphere promotes shallower cumuli because both the entrainment of moister air into the boundary layer, which decreases surface fluxes, and a weaker radiative

cooling at the cloud layer top lead to a weaker buoyancy excess in clouds compared to their environment and therefore convection remains shallower (e.g. Nuijens and Siebesma, 2019).





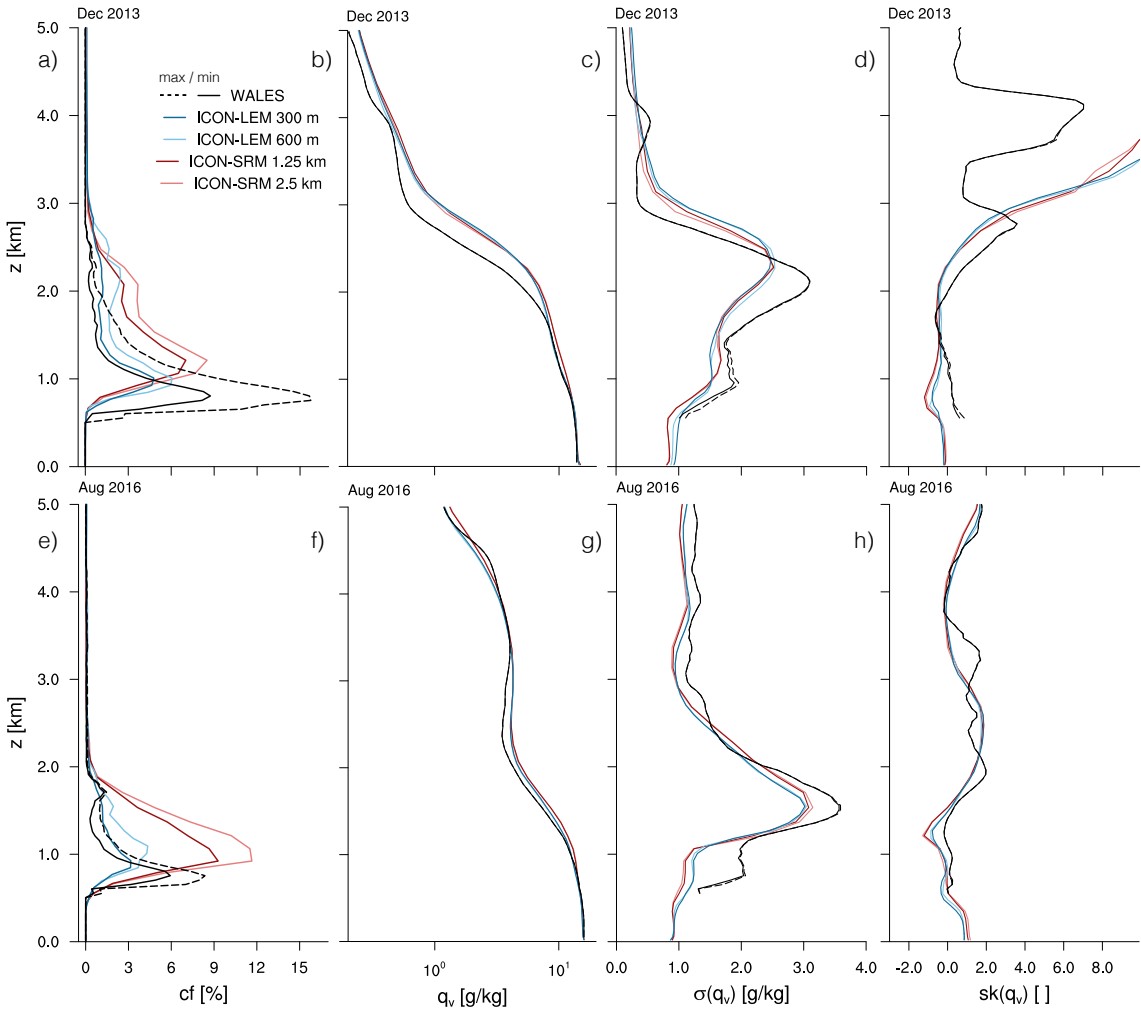

**Figure 8.** Profiles of (a,e) cloud fraction, (b,f) mean water vapor. $q_v$, and its (c,g) standard deviation and (d,h) skewness for the flight composites of (a-d) December 2013 and (e-h) August 2016 in stretched WVP space as defined in Fig. 7 b and c.

For two features there is better agreement between model and observations in the August composite than in the December composite: the moist model bias at the inversion is strongly reduced in August, and model and observations agree on a near-zero skewness of $q_v$ near cloud base. However, the upward shift in the modeled height of the maximum cloud fraction and the

underestimation of the standard deviation of $q_v$ in the cloud layer by the model remain. Compared to the SRM simulations at coarser resolution, the LEM simulations are better able to capture the height of the cloud maximum and the amount of cloud fraction except for the cloud base cloud fraction. The SRM simulations clearly overestimate the cloud fraction throughout the cloud layer above cloud base. Because cloud fraction is not converged in the LEM simulations, we expect an underestimation of cloud fraction as grid spacing approaches decameter scale.



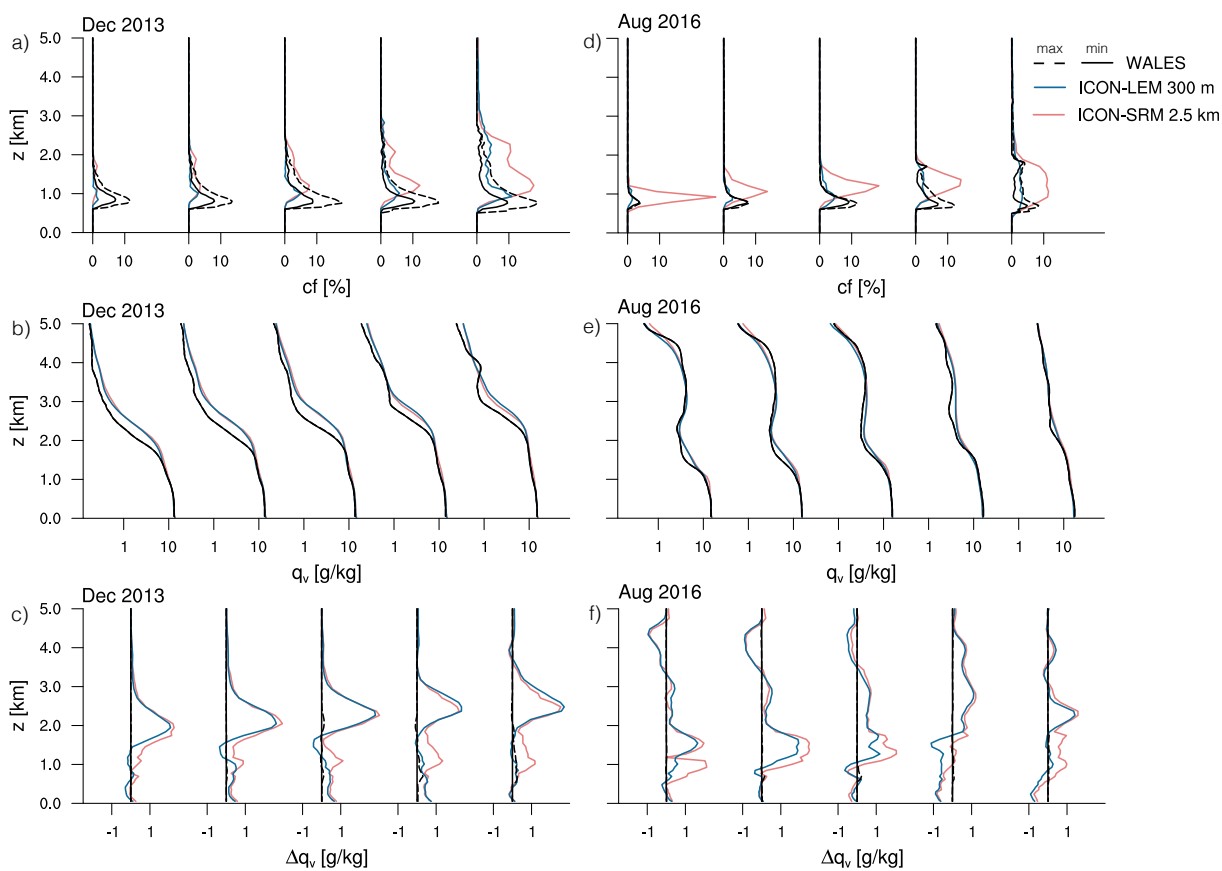

**Figure 9.** Profiles of (a,d) cloud fraction, (b,e) water vapor and (c,f) the difference to water vapor WALES$_{min}$ for flight composites (a-c) December 2013 and (d-f) August 2016. Each profile shows the mean for a 20-percentile range of WVP in stretched moisture space from driest profiles on the left to moistest profiles on the right (see Fig. 7).

A robust feature of the December and the August composite is the observed deepening of the cloud layer with increasing WVP from a few hundred meters for low WVP to the top of the inversion for high WVP (at 3 km in December 2013 and at 2 km in August 2016, Fig. 9 a, d). This deepening is well captured by the simulations across resolution.

     A better representation of cloud fraction with higher resolution becomes apparent for the covariation of cloud fraction with WVP. In the August composite the LEM simulations capture the observed increase in cloud fraction from cloud-free to about

10 % (Fig. 9 d). However, the transition from cloud-free to low cloud fractions occurs too late in moisture space in the LEM. In contrast the coarse resolution SRM simulates clearly too much cloud fraction in the driest part of the moisture space where none is observed and overestimates cloud fraction at high WVP throughout the cloud layer above cloud base. A too high low-cloud fraction increases the radiative cooling of the subcloud layer and can perhaps artificially promote convective self-aggregation too strongly when it is driven by low-level radiative cooling outside deep convective regions (e.g., Muller and Held, 2012;

Hohenegger and Stevens, 2016; Wing et al., 2017).





Different from the August composite, in the December composite even for the driest part of the moisture space a distinct cloud fraction is observed (Fig. 9 a). Neither the SRM nor the LEM are able to capture this cloud regime but instead simulate cloud-free conditions. While both observational estimates of cloud fraction agree well for the dry part of the moisture space, the picture is less clear for the moist part of the moisture space. For WALES$_\text{max}$ the maximum cloud fraction increases with

increasing WVP but for WALES$_\text{min}$ it is close to constant. The SRM and LES results both show increasing cloud fraction with increasing WVP but due to the uncertainty from the observational estimate, we cannot confirm this behaviour with WALES. Using ground based observations that are better able to estimate cloud fraction near cloud base, Nuijens et al. (2013) find that most of the variability in cloud fraction comes from clouds aloft and that clouds near the LCL are rather invariant with time. Although the variability depends on the time scale considered, this and the theory of the cumulus valve mechanism (Neggers

et al., 2006; Bellon and Stevens, 2013) seem to be supported by the WALES$_\text{min}$ estimate of a constant cloud fraction near cloud base in moisture space, but not by WALES$_\text{max}$.

Differences in the vertical distribution of water vapor between model and observations are more subtle than those in cloud fraction. The observed rate of increase in inversion height in moisture space is well captured by the simulations (Fig. 9 b,e). In both the model and the observations the increase in WVP is mostly accomplished by a deepening of the moist layer and to

a lesser extent by increasing moisture in the subcloud layer or above. If the increase in WVP was solely due to a deepening of the moist layer, then the agreement in the deepening rate between observations and simulations would directly follow from their agreement in percentile distribution of WVP (Fig. 7). It can therefore not be seen as a fully independent feature.

In the December composite the simulated inversion is shifted upward independent of WVP, which causes a strong bias around 2 km height (Fig. 9 c). For the December and the August composite the simulated gradients at the inversion are smoother than

those observed, a well-known difficulty of simulating inversions in particular if vertical resolution is moderate. (In ICON-LEM the vertical grid spacing is about 100 m at 2 km height, for ICON-SRM 200 m.) Because the gradient of moisture at the inversion plays an important role for the local radiative fluxes, the weaker gradient implies a less concentrated radiative cooling in the simulations at the cloud layer top. Besides the too high cloud fraction at kilometer-scale resolution discussed above, the too smooth moisture gradient at the inversion is another model feature that distorts the interaction between radiation,

subsidence and cloud development.

Model biases in $q_v$ also lead to misrepresentations in modeled cloud fraction. In the August composite in the driest 20 percentiles of moisture space, the SRM is too moist between 500 m and 1000 m that is where there is too high cloud fraction. For the mid-range percentiles of moisture space (between the 20 percentile and 60 percentile) the bias in modeled $q_v$ shows a bipolar structure for both SRM and LEM: On the one hand, at the height of the observed cloud maximum the modeled $q_v$

is slightly too low, coinciding with modeled spurious too low cloud fraction at the observed cloud base. On the other hand, around the inversion the modeled $q_v$ is too high, coinciding with spurious cloud fraction in the SRM at a height where there are much less clouds observed. We suspect that the latter feature only appears in the SRM simulation and not in the LEM simulation because the SRM applies a cloud fraction parameterization which can produce cloud cover at subsaturation. Taken together, the model smooths $q_v$ in the inversion and thereby puts moisture too high into the inversion region where it produces

clouds in the SRM and lacks moisture below the inversion where clouds are observed but not represented in the model.



## 5  Conclusions

In this study, we analyse the distribution of water vapor and clouds in the trades and how their covariation differs in observations and high-resolution models. The NARVAL campaigns, which took place in the northern tropical Atlantic east of Barbados, provide the opportunity to analyse the distribution of water vapor in the trade wind regime of shallow cumulus cloud fields

and in the vicinity of deep convection (Stevens et al., 2019b). In this study, we analyse five research flights from December 2013 probing the region's dry season and four research flights from August 2016 probing the region's moist season. With a horizontal resolution of 2.5 km, the WALES lidar during the NARVAL campaigns provides accurate measurements of the water vapor distributions primarily in the cloud-free gaps of the shallow cumulus regime. The lidar data are compared with results from nested ICON model runs that are available at four grid spacings from 2.5 km to 300 m and that include the area

and period of the flight domains.

Because of its stochastic nature, shallow convection is not expected to trigger at the exact same location and time in simulations as it does in reality. To bypass the issue of co-location but retain information on variability, we sort water vapor profiles from the driest to the wettest profile and compare simulations and observations in moisture space (Bretherton et al., 2005; Schulz and Stevens, 2018). Because the signal of the WALES lidar is attenuated rapidly when encountering a cloud and there-

fore preferentially misses cloudy, high moisture profiles, information from the HAMP radiometers co-located with the lidar is used to construct a "stretched moisture space" that enables a fair comparison between WALES and ICON.

Across model grid spacing from hecto- to kilometer scale, ICON is able to represent the observed features of the water vapor distribution well. In stretched moisture space it correctly captures the full range of WVP from 20 kg/m$^2$ to 55 kg/m$^2$, the vertical distribution of the first three moments of water vapor, and the variability of water vapor profiles across moisture

space. An exception in the vertical distribution is a persistent moist model bias at the trade wind inversion in the dry season. In both seasons the model tends to smooth the moisture gradient at the inversion too much, which is a known feature of excessive model diffusion and might also be a result of underresolving shallow convection with low horizontal resolution. In addition, the simulations slightly underestimate the variability of water vapor in the cloud and subcloud layer in both seasons. Both the too smooth inversion gradient and the too weak cloud layer variability are expected to distort the interaction between radiation,

subsidence and cloud development. That there is little dependence of these features on grid spacing and the general good agreement with observations implies no advantage of hectometer grid spacing over kilometer grid spacing in representing the water vapor distribution in the trade wind regime.

In contrast to water vapor, the modelled cloud fraction strongly depends on grid spacing. While the observed cloud deepening with increasing moisture is captured well across model resolutions, the modeled cloud fraction strongly decreases with

increasing grid resolution. In the dry season the observational uncertainty in cloud fraction is too large to make a firm statement. In the wet season simulations with hectometer grid spacing agree better with observations than simulations with kilometer grid spacing. In particular, the transition from cloud-free to low cloud fraction with increasing moisture, which reflects the close connection between the distribution of water vapor and clouds, is better represented at hectometer resolution. Also, the height of maximum cloud fraction, which is observed just above cloud base, is shifted upward in the model in both seasons but de-





creases with higher resolution towards the observed values. Although cloud amount and its vertical distribution is compelling at
300 m grid spacing, it is not converged yet, which is in line with idealized modelling studies showing that LEM underestimates
cloud fraction for decameter grid spacing (Vogel et al., 2019).

In conclusion, we show that high-resolution simulations of the shallow cumulus trade wind regime with kilometer scale grid
spacing and realistic boundary conditions are able to capture the characteristics of the lower tropospheric water vapor distribu-
tion well (Heinze et al., 2017; Stevens et al., 2019a). They however have difficulties to reproduce the observed covariation of
water vapor and cloud statistics, which is improved at hectometer resolution. As has been shown for conventional climate mod-
els, which apply a convective parameterization at much coarser resolution (e.g., Jiang et al., 2012), this means that capturing
the water vapor distribution correctly does not imply that shallow clouds that live at the tail of the water vapor distribution are
also well represented. It remains an open question which role such shallow cloud biases in kilometer-scale simulations play for
the heat budget of the cloud layer and how they interact with the large-scale environment, e.g., in global storm resolving models
(Satoh et al., 2019). The latter question of whether and how shallow cloud biases depend on the large-scale environment also
prompts itself to be pursued further in the light of EUREC[4]A, which sets out for measuring the distribution of water vapor and
clouds in conjunction with the large-scale environment (Bony et al., 2017).

*Code and data availability.* Model results and observational data used in this study are published in different peer reviewed papers, that is
ICON-SRM NARVAL 1+2: Klocke et al. (2017); ICON-LEM NARVAL 1+2: Stevens et al. (2019b); WALES NARVAL 1: Kiemle et al.
(2017); WALES NARVAL 2: Gutleben et al. (2019); HAMP NARVAL 1+2: Jacob et al. (2019c, a, b).

*Author contributions.* AKN and CK developed the idea of the study and carried out the analysis for this manuscript. AKN took the leading
role in writing the manuscript with input from CK.

*Competing interests.* The authors declare that they have no conflict of interest.

*Acknowledgements.* We thank Bjorn Stevens for inspiring discussions and in particular for his idea to include radiometer data in this study.
We also thank Marek Jacob for kindly providing the HAMP data, Silke Gross and Martin Wirth for the WALES data, and Daniel Klocke
and Matthias Brück for the ICON results used in this study. The data used in this publication was gathered in the NARVAL 1/NARVAL 2
campaigns and WALES data is made available through the German Aerospace Center (DLR). NARVAL was funded with support of the Max
Planck Society, the German Research Foundation (DFG, project HALO-SPP 1294), the European Research Council (ERC), the German
Meteorological Weather Service (DWD) and DLR. Primary data and scripts used in the analysis and other supplementary information that
may be useful in reproducing the author's work are archived by the Max Planck Institute for Meteorology and can be obtained by contacting
publications@mpimet.mpg.de. During part of this research AKN was funded as part of the Hans-Ertel Centre for Weather Research. This



research network of Universities, Research Institutes and the Deutscher Wetterdienst is funded by the BMVI (Federal Ministry of Transport and Digital Infrastructure). AKN also acknowledges funding by the Deutsche Forschungsgemeinschaft (DFG, German Research Foundation)

under Germany's Excellence Strategy – EXC 2037 'Climate, Climatic Change, and Society' – Project Number: 390683824.



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
