# Peer review of "The vertical structure and spatial variability of lower tropospheric water vapor and clouds in the trades"

_Atmospheric Chemistry and Physics, 2019_

## Referee Comment (RC1) · Anonymous Referee #1 · 22 Jan 2020

GENERAL COMMENTS

- Despite its importance many questions on water vapor characteristics are still poorly understood. The paper uses unique water vapor profile measurements from two campaigns in the North Atlantic trades to investigates how well numerical simulations at different resolution capture the water vapor variability and its subsequent impact on cloudiness. This topic as well as the evaluation metrics used to investigate the problem are interesting and innovative making the paper well suited for ACP.

- Water vapor variability includes both spatial and temporal changes acting on different scales. The paper does not explicitly discuss the scale of variability addressed by its

observations and the models and whether the same scales are captured. However, this is important as variability itself depends on the considered as shown by Steinke et al. (2015, www.atmos-chem-phys.net/15/2675/2015/, their fig. 4) for a convective boundary layer with ICON. Due to the nature of the airborne measurements it is not possible to disentangle spatial and temporal variations from the observations but this could be done in the model world. In fact in the observations I suspect that there will be more correlation as in the model as spatially neighbouring profiles are correlated while model profiles are randomly distributed. Maybe this issue can be assessed by checking different approaches to select the model data, e.g. along straight lines resembling flight paths?

- The water variability assessed by the paper (with 2.5 km grid size for ICON-SRM) is not on the same scale as the shallow clouds which have typically much shorter dimensions. There should be some information on cloud dimensions available from lidar or other measurements. The issue needs to be discussed and might be addressed in a follow-up analysis taking also cloud length into account which could be derived from backscatter lidar.

- When trying to connect water vapor and clouds it is also interesting to look at the water budget. Water vapor mixing ratio qv might not differ much for the max and min scenarios but this difference is likely in the order of the liquid water vapor mixing ratio qc. Therefore checking how this difference translates in the end to cloud fraction might give new insights as also the microwave radiometer should be able to provide LWP simultaneously with WVP. This analysis could support the conclusion that water vapor variability not necessarily needs to relate to an adequate representation of clouds as these live at the tail of the water vapor distribution. In this respect it is interesting to know how strong temperature variability is? Would it be possible to look at relative humidity?

- Being old fashion and looking at a printout several figures are very difficult to read and I make several suggestions for improvements in the technical section

SPECIFIC COMMENTS

L7 please avoid the term "humidity inversion" as this would point at the classic polar phenomena of increasing moisture with altitude which is not what you mean. Also in line 94 this should be rephrased to make clear that you talk about the temperature inversion.

L8 "but is less pronounced" than what?

L28 "with the decrease in subsaturation in the column" Is this true for all seasons. If you compare the different degrees of saturation in the free troposphere during the wet and dry season?

L27 "profiling moisture, aerosol, and clouds simultaneously with high accuracy and spatial resolution" is a bit overselling as there are limitations set by the strong lidar attenuation by clouds

L115 For LCL it would be good to say how good the lidar approach is compared to drop sonde profiles?

L133-135 I do not understand this sentence. How do you know that 1.5 % of the radiometer WVP data are affected by "saturation"?

L136: ICON has a complex grid such that resolution is not exactly the grid size, however, the true resolution of a model will always be coarser than the grid size. A discussion is needed.

L153 SST fixed for each simulation day. However, SST shows spatio-temporal variation. Does SST have an influence on water vapor variability or cloud fraction?

L184: Fig. 3 combines spatial and temporal variability. I would be good to split this up and check which limitation is imposed by the individual contributions. Just assuming a classical 10 m/s advection time scale gives an equivalent scale of 36 km for one hour time (ICON output frequency). That is of course a simplified view but could easily

explain why models with scale 20 km and below are so similar.

L210-218: Are the numbers given here for the whole campaign or as in Fig. 4 only for 11 Dec 2013? In fact it might be good to explain before how the stretched WVP scale is generated for days and campaign?

L329: Any idea why not? Are they to optically thick and thus as stratiform layers cover larger scales not in the data set?

L348: The paper shows the better representation of cloud fraction for the "higher resolution" simulation. This does not necessarily need to be a resolution effect but might be due to the different cloud schemes employed by the SRM and LEM. One possibility might be autoconversion which might be too weak in SRM allowing further vertical development of the clouds.

L395:" in the vicinity of deep convection" is a significant part of the data from this region?

L409: "..the major features of the vertical distribution"

TECHNICAL CORRECTIONS

Fig.1 is perfectly suited to add a fourth subplot with the difference between WVP_max-WVP_min which I find missing.

Fig. 2: I can't see anything in the water vapor plots. Maybe add a few contour lines. Wouldn't it make sense to show a microwave satellite field for WVP (from SSM/I, AMSR..)?

Fig. 4: I can't distinguish the different lines. Anyhow Fig. 4a already nicely shows both WVP scale so that I think that 4b could better show the difference of the models to values instead of repeating the full scale.

Fig. 7: similar to fig. 4 her b and c should be plotted as anomalies.

[Figure]

---

## Referee Comment (RC2) · Anonymous Referee #3 · 19 Feb 2020

Review of "The vertical Structure and spatial Variability of lower tropospheric Water Vapor and Clouds in the Trades" (acp-2019-1015) by A.K. Naumann and C. Kiemle

Synopsis

The present study investigates the variability of water vapor and clouds in the tropical Atlantic. The manuscript is mainly a model validation, i.e., it focuses on the comparison of observations with simulations that have different grid spacings. The authors found that the variability of water vapor is generally well represented by the simulations with little differences between the various model resolutions, whereas the simulated cloud field shows a stark dependence on model resolution.

[Figure]
* * *
Interactive
comment

Overall comments

Overall, I think this study is appropriate for publication in ACP. I do not see any major flaws. The approach is sound, the results are sound, and, with a few exceptions detailed below, follow from the evidence. I do think, though, that the whole manuscript reads a little bit tedious; in other words, the clarity of the writing could be improved. Sometimes I feel like the authors make things more complicated than they really are. Examples are given in the specific comments below.

Specific comments

- I like that the authors make an effort to quantify uncertainty in the WVP measurements. This is really helpful in assessing potential model biases.

- I think that the entire case study (Section 3) could be removed without diluting the main points of the manuscript. The results aggregated over Dec. 2013 show a similar story, and the case study is not necessary to understand the aggregated results.

- I am still unclear what "spanning the moisture space" really means. Is it just the ordering of all individual profiles with respect to their WVP? Also, it is not fully clear to me what the "stretched moisture space" accomplishes.

- It seems like the WALES instrument has some issues with sensing water vapor in cloudy/very moist areas that HAMPS does not have. What's the reason for using WALES in conduction with HAMPS instead of HAMPS alone?

- How is the "cloud layer" defined?

- How are the model fields subsampled? Are model soundings drawn from under a virtual flight track?

- Text-figure mismatch hinders readability: For the multi-panel figures like. Fig. 5, the authors first describe panel b), then c) and d), and lastly a). Making the text and figure panel order consistent would improve clarity.

[Figure]

- paragraph beginning on line 183: I can't quite relate the text here to Fig. 3. Also, I'm not sure what Fig. 3 is supposed to explain.

- l. 252: "standard deviation of qv": How do you compute the standard deviation, in space or in time?

- l. 275: "...but is still included in the range of uncertainty given by the observations." — data from LEM 300m seem to fall outside the range estimated by WALES.

- l. 325: "a moist model bias at the inversion" — I think what's going on is that the simulated inversion is too high. This is equivalent to what you write but more intuitive when looking at the figure.

- l. 352: I find the sentence construction "A too high low-cloud fraction" difficult to follow, because is has words that are the exact opposite of each other...I suggest rephrasing.

- l. 410-411: "In both seasons the model tends to smooth the moisture gradient at the inversion too much, ..." — I'd say the more important bias is that the moist layer is too deep, or in other words, the inversion is too high in the models.

Technicalities, typos, etc.

l. 106/107: "and only in 34 % of all lidar profiles are more than half of the data points valid below this height" l. 123: What is the 12-s grid? I don't think this has been mentioned before.

---

## Author Comment (AC1) · 7 Apr 2020

Please find our reply to the reviewer's comments and a revised manuscript with trackable changes in the supplement.

Please also note the supplement to this comment:
https://www.atmos-chem-phys-discuss.net/acp-2019-1015/acp-2019-1015-AC1-supplement.pdf

---

## Author Response (AR1)

**Reply to reviewers: The vertical structure and spatial variability of lower tropospheric water vapor and clouds in the trades**

Ann Kristin Naumann, Christoph Kiemle

We thank the reviewers for their helpful comments on the manuscript. In the following, reviewer's comments are in *italics*, authors' responses are in normal font.

**Anonymous Referee 1**

*GENERAL COMMENTS*

*Despite its importance many questions on water vapor characteristics are still poorly understood. The paper uses unique water vapor profile measurements from two campaigns in the North Atlantic trades to investigates how well numerical simulations at different resolution capture the water vapor variability and its subsequent impact on cloudiness. This topic as well as the evaluation metrics used to investigate the problem are interesting and innovative making the paper well suited for ACP.*

We thank the reviewer for the positive assessment.

*Water vapor variability includes both spatial and temporal changes acting on different scales. The paper does not explicitly discuss the scale of variability addressed by its observations and the models and whether the same scales are captured. However, this is important as variability itself depends on the considered [scale (?)] as shown by Steinke et al. (2015, www.atmos-chem-phys.net/15/2675/2015/, their fig. 4) for a convective boundary layer with ICON. Due to the nature of the airborne measurements it is not possible to disentangle spatial and temporal variations from the observations but this could be done in the model world. In fact in the observations I suspect that there will be more correlation as in the model as spatially neighbouring profiles are correlated while model profiles are randomly distributed. Maybe this issue can be assessed by checking different approaches to select the model data, e.g. along straight lines resembling flightpaths?*

We agree that it is not possible to untangle spatial and temporal variability from the observations. For the simulations, the variability of each day is clearly dominated by the spatial variability. This can be seen in Fig. 1 in this document, where we show that the standard deviation of $q_v$ does not differ considerably if we select one of the hourly output time steps (blue lines) or the full period (black solid line).

We also agree that we would expect different correlation scales for line-like observations than in

[Figure]

Figure 1: Standard deviation of $q_v$ on 11 December 2013 from ICON-SRM 2.5 km for the full flight domain and period and for different sampling strategies.

the randomly selected profiles. However, for the variability the ordering does not play a role: if we select 10 latitudinal lines to mimic the flight path (dashed black line) or randomly sample 25 % of the profiles (red dashed line), the standard deviation of $q_v$ does not differ considerably from the standard deviation of the full domain (solid black line). Only if we sample for the stretched WVP space, the standard deviation decreases because the sampling for the stretched WVP space is biased to where the lidar can measure, i.e., to outside of clouds where it is typically drier. We believe that the temporal variability plays a much smaller role in our analysis than for Steinkeel et al (2015) because of the steadiness of the trade wind regime compared to the midlatitudes. Also, the relatively large domain size might play a role.

We hesitate to add the figure to the manuscript because the direct comparison to the observations is not possible and because an in depth discussion of the temporal and spatial variability is beyond the scope of the manuscript. However, we add a clarification and short discussion of the topic in l. 159: "We analyse all model output in these domains instead of selecting profiles along the flight tracks because convection is not expected to trigger at the exact same location and time in simulations as it does in reality. Using the domain output is consistent with the statistical rather than spatio-temporal approach of this analysis and promotes the robustness of the results." and in line 197: "The analysis combines spatial and temporal variability but the contribution from spatial variability is dominating (not shown)."

*The water variability assessed by the paper (with 2.5 km grid size for ICON-SRM) is not on the same scale as the shallow clouds which have typically much shorter dimensions. There should be some information on cloud dimensions available from lidar or other measurements. The issue*

*needs to be discussed and might be addressed in a follow-up analysis taking also cloud length into account which could be derived from backscatter lidar.*

Cloud size information from the lidar is provided in Gutleben et al. (2019). Using all flights from the two NARVAL campaigns they find that around two thirds of all trade wind clouds have a horizontal extent of less than 0.5 km. However, the contribution of these small clouds to overall cloud fraction scales with the size of the clouds. In ASTER satellite images of typical trade wind regimes (Mieslinger et al, 2019), clouds with a horizontal extent of less than 350 m contribute less than 10 % to the overall cloud fraction. We add this discussion in l. 282: "Typical cloud sizes obtained from the lidar are around 500 m (Gutleben et al 2019) and hence on the order of the grid spacing of the simulations. Because the contribution to overall cloud fraction scales with the size of the clouds, we do not expect the contribution of these small clouds to dominate the overall cloud fraction."

*When trying to connect water vapor and clouds it is also interesting to look at the water budget. Water vapor mixing ratio qv might not differ much for the max and min scenarios but this difference is likely in the order of the liquid water vapor mixing ratio qc. Therefore checking how this difference translates in the end to cloud fraction might give new insights as also the microwave radiometer should be able to provide LWP simultaneously with WVP. This analysis could support the conclusion that water vapor variability not necessarily needs to relate to an adequate representation of clouds as these live at the tail of the water vapor distribution. In this respect it is interesting to know how strong temperature variability is? Would it be possible to look at relative humidity?*

Here care is required in order not to confuse different things: the difference between the min and max estimate from the lidar quantifies the measurement uncertainty in the WVP due to the inability of the lidar to penetrate clouds. However, one might ask how the bias in the modeled WVP translates into cloud fraction from an order-of-magnitude water budget perspective. This is not an easy task either: Typical $q_v$ values in the cloud layer ($10$ g kg$^{-1}$) are a factor of 30 larger than the in-cloud $q_c$ ($0.3$ g $kg^{-1}$). From a water budget perspective, were the bias in WVP (Fig. 7 in the manuscript, say 1.2 kg m-2) not concentrated near the inversion but instead fully converted into liquid water and then distributed over the cloud layer depth (say 2000 m) with a typical in-cloud $q_c$, this would easily turn the whole column cloudy twice. However, we believe this estimate is flawed because the cloud layer on average is not saturated but typical relative humidities are below 90 %. As the other extreme, one could distribute all the WVP bias over the cloud layer in a way that it only increased the relative humidity without any additional saturation anywhere and hence no increase in cloud fraction. Due to the large range obtained by these two thought experiments, we believe such an order-of-magnitude argument is unfortunately not helpful at this point.

To look at relative humidity, one needs an estimate of the temperature profile, which is available from dropsondes but they are much more sparse in time and space than the lidar measurements. To avoid introducing this additional uncertainty, we concentrate on the specific humidity instead of relative humidity in the manuscript. In principle also the LWP is available from the microwave

radiometer but because an order-of-magnitude budget analysis seem not to be useful in this context, we decided to focus on the in-depth analysis on the vapor phase and specific humidity for this study.

*Being old fashion and looking at a printout several figures are very difficult to read and I make several suggestions for improvements in the technical section*

Thanks for the comments. We mostly followed the suggestions and adjusted Fig. 1, 2, 4 and 7. Please see our answers in the technical section.

*SPECIFIC COMMENTS*

*L7 please avoid the term "humidity inversion" as this would point at the classic polar phenomena of increasing moisture with altitude which is not what you mean. Also in line 94 this should be rephrased to make clear that you talk about the temperature inversion.*

In line 7 we reformulate "... too weak humidity gradient at the inversion near the cloud top." and in line 95 "... the inversion that tops the cloud layer in the trades... ".

*L8 "but is less pronounced" than what?*

To clarify we added "... less pronounced than the moist model bias at the inversion".

*L28 "with the decrease in subsaturation in the column" Is this true for all seasons. If you compare the different degrees of saturation in the free troposphere during the wet and dry season?*

To our understanding this has been shown for all seasons in the deep convective regions. For shallow convection there are less studies but Nuijens et al (2009) show that it is true at least within the dry season in the trades, where they say that "... most of the variability in the humidity profiles, when conditioned on precipitation, is in the lower free troposphere and little is in the boundary layer ...". We clarify that in the manuscript l. 28: "... the amount of precipitation in deep convective regions correlates well with the decrease in subsaturation in the column (Bretherton et al., 2004; Holloway and Neelin, 2009). The same relation is found to hold within the dry season in the shallow convective regime (Nuijens et al., 2009)."

*L27 "profiling moisture, aerosol, and clouds simultaneously with high accuracy and spatial resolution" is a bit overselling as there are limitations set by the strong lidar attenuation by clouds*

We agree that the lidar is not able to profile clouds due to attenuation but rather able to detect cloud tops. We changed the text accordingly: "... profiling moisture and aerosols, and detecting cloud tops simultaneously with high accuracy and spatial resolution".

*L116 For LCL it would be good to say how good the lidar approach is compared to dropsonde profiles?*

We have tried to develop an algorithm to automatically retrieve the LCL from the lidar signals using cloud base heights detected from thin clouds the lidar can penetrate. These mostly correspond well with the LCL derived from the dropsonde profiles, however, due to the inability to detect the base of thick clouds with lidar the results from such an approach are slightly biased. Therefore we preferred a comprehensive case-to-case analysis of each situation, including the dropsonde results,

together with auxiliary lidar information such as the aerosol and water vapor gradients at the top of the MBL. We added this indeed important information to the manuscript in line 120: "To find the LCL, we use the lidar signals from thin boundary layer clouds as well as dropsonde profiles and auxiliary lidar information such as aerosol and water vapor gradients at the top of the mixed layer. "

*L133-135 I do not understand this sentence. How do you know that 1.5 % of the radiometer WVP data are affected by "saturation"?*

Thanks for pointing this out. In this context, "saturation" is ambiguous and we replace it by "signal attenuation" in the manuscript.

*L136: ICON has a complex grid such that resolution is not exactly the grid size, however, the true resolution of a model will always be coarser than the grid size. A discussion is needed.*

The nominal grid spacing of the spherical icosahedron ICON grid is discussed by Wan et al. 2013 and Giorgetta et al. 2018. The effective resolution of ICON LEM is estimated to be a factor of six to ten larger than the nominal grid spacing by Hansen et al. 2020. We add this in the manuscript.

*L153 SST fixed for each simulation day. However, SST shows spatio-temporal variation. Does SST have an influence on water vapor variability or cloud fraction?*

Spatial variation of SST are included in the simulation setup. SST also differs from one simulation day to the other but not temporally within one simulation (which in our case is done for 36 h lead time). Ocean-atmosphere coupling and its influence on clouds in the trades is a topic of ongoing research (e.g., in the atmosphere-ocean coupling component of EUREC4A; Bony et al 2017). To our understanding in the tropics the effect of temporal variations of SST on the time scale of less than 2 days is most pronounced for tropical cyclones. For the trades, Vial et al (2019) show that the diurnal cycle of cloudiness is well represented, even if the SST is fixed for 36 h.

*L184: Fig. 3 combines spatial and temporal variability. I would be good to split this up and check which limitation is imposed by the individual contributions. Just assuming a classical 10 m/s advection time scale gives an equivalent scale of 36 km for one hour time (ICON output frequency). That is of course a simplified view but could easily explain why models with scale 20 km and below are so similar.*

Please see our answer to the first comment and Fig. 1 in this document.

*L210-218: Are the numbers given here for the whole campaign or as in Fig. 4 only for 11 Dec 2013? In fact it might be good to explain before how the stretched WVP scale is generated for days and campaign?*

The numbers given here are for 11 Dec 2013 and we added a clarification in the manuscript. Because all of Section 3 is about the case study, we fear that might be confusing to the reader to talk about the stretched WVP space of the seasonal composites at this point. Instead we clarified the peculiarities of the seasonal composite in Section 4 l. 310: "As for the case study in the previous section, we subsample all model results according to the percentages available from WALES in each 10 % bin of WVP for each flight individually. After the subsampling we

concatenate the individual flights into the seasonal composite. The composite is thus weighted by the number of valid profiles per flight (which vary from flight to flight; Table 1).".

*L329: Any idea why not? Are they to optically thick and thus as stratiform layers cover larger scales not in the data set?*

Thanks for the comment. We checked the original unfiltered lidar data: Most of the stratiform cloud layers are indeed large and too optically thick, and hence not present in the analysed data set. We added this in the manuscript, l. 349: "... but are mostly removed from our analysis of the WALES data due to their opacity."

*L348: The paper shows the better representation of cloud fraction for the "higher resolution" simulation. This does not necessarily need to be a resolution effect but might be due to the different cloud schemes employed by the SRM and LEM. One possibility might be autoconversion which might be too weak in SRM allowing further vertical development of the clouds.*

Thank you for the suggestion. The role of the autoconversion rate has recently been highlighted by Jacob et al (2020). We include this in the discussion of Section 3, l. 289: "Another hypothesis, which has recently been developed by Jacob et al (2020), proposes that slight differences in the parameterization of autoconversion in the SRM and LEM might cause differences in the cloud's vertical extent."

*L395: "in the vicinity of deep convection" is a significant part of the data from this region?*

The majority of measurements are from the trade wind regime, hence we deleted the phrase here.

*L409: "..the major features of the vertical distribution"*

We added that.

*TECHNICAL CORRECTIONS*

*Fig.1 is perfectly suited to add a fourth subplot with the difference between WVPmax-WVPmin which I find missing.*

Thanks for the suggestion. We added a fourth plot with the WVP from WALES$_{max}$, WVP$_{min}$ und HAMP and $\Delta$WVP to Fig. 1.

*Fig. 2: I cant see anything in the water vapor plots. Maybe add a few contourlines. Wouldnt it make sense to show a microwave satellite field for WVP (from SSM/I,AMSR..)?*

Unfortunately, contour lines don't work well for the ICON-LEM WVP because of the fine-scale structures at 300 m grid spacing. Instead we adjusted the scale from originally 20 - 60 kg m-2 to now 25 - 45 kg m-2, so that the contrast shows stronger now. For the WVP, we prefer to not add satellite data to keep the study focused on the comparison between lidar and simulations.

*Fig. 4: I cant distinguish the different lines. Anyhow Fig. 4a already nicely shows both WVP scale so that I think that 4b could better show the difference of the models to values instead of repeating the full scale.*

Fig. 4b differs from Fig. 4a by the applied subsampling. We think it is worthwhile to illustrate this

transformation from "WVP space" to the "stretched WVP space" because it is a key method of the analysis and might not be very intuitive for the reader. However, we agree that the differences of the simulations to the measured values are also interesting but difficult to see in Fig. 4b. We therefore added an anomaly plot as Fig. 4c.

*Fig. 7: similar to fig. 4 her b and c should be plotted as anomalies.*

As for Fig. 4, we added the anomalies as Fig. 7 d and e.

**Anonymous Referee 3**

*Synopsis: The present study investigates the variability of water vapor and clouds in the tropical Atlantic. The manuscript is mainly a model validation, i.e., it focuses on the comparison of observations with simulations that have different grid spacings. The authors found that the variability of water vapor is generally well represented by the simulations with little differences between the various model resolutions, whereas the simulated cloudfield shows a stark dependence on model resolution.*

*Overall comments: Overall, I think this study is appropriate for publication in ACP. I do not see any major flaws. The approach is sound, the results are sound, and, with a few exceptions detailed below, follow from the evidence. I do think, though, that the whole manuscript reads a little bit tedious; in other words, the clarity of the writing could be improved. Sometimes I feel like the authors make things more complicated than they really are. Examples are given in the specific comments below.*

We followed the reviewers suggestions in the specific comments, please see our replies below.

*Specific comments*

*I like that the authors make an effort to quantify uncertainty in the WVP measurements. This is really helpful in assessing potential model biases.*

Thank you.

*I think that the entire case study (Section 3) could be removed without diluting the main points of the manuscript. The results aggregated over Dec. 2013 show a similar story, and the case study is not necessary to understand the aggregated results.*

We perform the case study in Section 3 for two main reasons: First, we think it is helpful for a reader to get an impression of the data, see its limitation and understand how it is connected to the synoptic situation. This would be repetitive to do for all research flights, so we decided to present a single day in more detail in the form of a case study. Second, a key point of the manuscript is how the sampled lidar data and the simulation output then translate into moisture space and how adequate subsampling translates moisture space into stretched moisture space. Since this is a new method, we strive to explain the method in detail in the relatively homogenous dataset of a single research flight. Untangling these aspects in an aggregated seasonal composite is much harder. The reviewer's next comment seems to support our notion that a careful presentation of

the method is required.

We try to motivate the presentation of the case study more clearly in the introduction of Section 3, l. 173: "In this section, we use one day of the first NARVAL campaign, 11. December 2013, for a detailed case study. The aim of the case study is to introduce the central method of this study: the concept of a stretched moisture space. The stretched moisture space is obtained by selective subsampling of the model results and thereby allows for a fair comparison between lidar data and model results. The case study also illustrates some prominent features of covariation of clouds and moisture, before aggregated seasonal composite enable us to generalize the results to different regimes of water vapor structure in the trades in Section 4."

*I am still unclear what "spanning the moisture space" really means. Is it just the ordering of all individual profiles with respect to their WVP? Also, it is not fully clear to me what the "stretched moisture space" accomplishes.*

Correct, with moisture space we refer to an ordering of individual profiles with respect to their WVP value. This concept has been introduced by Bretherton et al (2005) and applied to observations by Schulz and Stevens (2018). Because the lidar quickly attenuates in clouds, its profiles are biased towards dry profiles (see next comment) and cannot be directly compared to simulation results. In stretched moisture space, we deliberately bias the sample of simulated profiles in the same way as the lidar profiles are biased. Therefore a fair comparison is accomplished only in stretched moisture space.

We explain the concept of moisture space and the motivation for comparing observations and simulations in stretched moisture space in detail in the first two paragraphs of Section 3.

*It seems like the WALES instrument has some issues with sensing water vapor in cloudy/very moist areas that HAMPS does not have. Whats the reason for using WALES in conduction with HAMPS instead of HAMPS alone?*

The HAMP radiometer measurements lack vertical profile information. Since most of our results are based on the analysis of vertical profiles, we use HAMP both as a cross check for WALES and also, and foremost, as an indicator of the WALES limits for high WVPs. We clarify this in line 135: "The nadir-viewing HAMP microwave radiometers lack vertical profile information but measure the WVP with 1 s (that is 210 m or 240 m) resolution along the HALO flight track also in the presence of shallow clouds (Jacob et al. 2019)."

*How is the "cloud layer" defined?*

We clarify that in the caption of Fig. 3: "The cloud layer ranges from cloud base at $z = 0.5$ km to the highest cloud tops at $z = 3.0$ km."

*How are the model fields subsampled? Are model soundings drawn from under a virtual flight track?*

Concerning the subsampling, please see our comment on the reviewer's question to the "stretched moisture space". The model soundings are drawn from the full domains given in Tab. 1. We clarify that in l. 159: "We analyse all model output in these domains instead of selecting profiles

along the flight tracks because convection is not expected to trigger at the exact same location and time in simulations as it does in reality. Using the domain output is consistent with the statistical rather than spatio-temporal approach of this analysis and promotes the robustness of the results."

*Text-figure mismatch hinders readability: For the multi-panel figures like Fig. 5, the authors first describe panel b), then c) and d), and lastly a). Making the text and figure panel order consistent would improve clarity.*

Because many scientist first look at the figures of a paper before they read (or even might not read) a paper, we ordered the panels of all figures in the way that we think is most easy to understand. For Fig. 5 we think it is useful to first look at the cloud fraction in (a), which best gives an impression of the analysed regime. This sets the context for the mean, standard deviation and skewness of the humidity profiles in (b), (c) and (d). When reading the full manuscript, the reader already knows about the cloud regime and we find it reasonable to first discuss the humidity profiles and then the cloud profiles that live at the tail of the humidity distribution. To help the reader navigate, we always refer to the individual panel of each figure (e.g., Fig. 5 b).

*paragraph beginning on line 183: I cant quite relate the text here to Fig. 3. Also, Im not sure what Fig. 3 is supposed to explain.*

We clarified this in line 196 in the manuscript: "Averaging the results of the ICON-LEM 300 m simulation on squares of different side length, we analyse how the standard deviation of the water vapor mixing ratio, $q_v$, depends on the considered scales (Fig. 3)." Because the simulations and the lidar have different native scales, it is not obvious whether they need to be averaged to a common resolution for a fair comparison. The small differences between humidity variability on scales of 300 m and 20 km in Fig. 3, allow us to compare the different dataset at their native resolution without the need to artificially reduce information by averaging.

*l. 252: "standard deviation of qv": How do you compute the standard deviation, in space or in time?*

Please see our answer to the first reviewer's first comment. We add a short discussion in the manuscript in line 197: "The analysis combines spatial and temporal variability but the contribution from spatial variability is dominating (not shown)."

*l. 275: "...but is still included in the range of uncertainty given by the observations." data from LEM 300m seem to fall outside the range estimated by WALES.*

Given all the uncertainties that come with such a comparison we think that 5.8 % (ICON LEM 300 m) is still too close to 7.4 % (WALES$_{min}$) to be considered substantially different. However, we agree that it is not strictly "in the range" and change the phrase to "... but is still close to the range of uncertainty given by the observations".

*l. 325: "a moist model bias at the inversion" I think whats going on is that the simulated inversion is too high. This is equivalent to what you write but more intuitive when looking at the figure.*

We replace the phrase in the manuscript by "a too high model inversion".

*l. 352: I find the sentence construction "A too high low-cloud fraction" difficult to follow, because is has words that are the exact opposite of each other...I suggest rephrasing.*

We rephrase: "If the low-cloud fraction is too large..."

*l. 410-411: "In both seasons the model tends to smooth the moisture gradient at the inversion too much,..." Id say the more important bias is that the moist layer is too deep, or in other words, the inversion is too high in the models.*

We rephrased the sentence before to stress the too high inversion in the model. This sentence, however, points to a second aspect.

*Technicalities, typos, etc. l. 106/107: "and only in 34 % of all lidar profiles are more than half of the data points valid below this height"*

We corrected that.

*l. 123: What is the 12-s grid? I dont think this has been mentioned before.*

The 12-s grid is introduced in line 102 but we reformulate the sentence in l. 132 to make it self-contained.

[revised manuscript text omitted]